# Influence of Fertilization on Growth and Lead Content of Pepper under Lead Stress

**DOI:** 10.3390/plants12162960

**Published:** 2023-08-16

**Authors:** Bingkun Yu, Dongying Xu, Yang Li, Wenquan Wang

**Affiliations:** College of Resources and Environment Sciences, Xinjiang Agricultural University, Urumqi 830052, China; youth6665989@foxmail.com (B.Y.); 15041865428@163.com (D.X.); radishlove@126.com (Y.L.)

**Keywords:** fertilization, Pb, pepper, stress, Urumqi

## Abstract

To investigate the effect of fertilization on Pb content in vegetables, pepper was planted in L_16_4^5^ (the 5 influencing factors are fertilizers (N, P, K), organic fertilizers (sheep manure) and Pb^2+^; the 4 levels are blank, low, medium and high; a total of 16 treatments) pot orthogonal experiment. The effects of fertilizers on the growth and Pb content in various parts of pepper under Pb stress were analyzed. The results showed that: (1) The Pb content in pepper fruit ranged from 0.011 mg·kg^−1^ to 0.085 mg·kg^−1^, which did not exceed the limit value (0.1 mg·kg^−1^) in the National Standard for Food Safety-Limit of Contaminants in Food (GB2762-2017); (2) The effect order of fertilization on pepper fruit weight was P_2_O_5_ > sheep manure > N > K_2_O; The horizontal combination of factors that promoted the maximum fruit weight of pepper was N (0.15 g·kg^−1^), P_2_O_5_ (0.225 g·kg^−1^), K_2_O (0.15 g·kg^−1^) and sheep manure (9 g·kg^−1^); (3) The order of fertilizer effects on Pb content in pepper fruit was Pb^2+^ > K_2_O > N = sheep manure > P_2_O_5_; the factor level combination that resulted in the maximum Pb content in pepper fruits was N (0.15 g·kg^−1^), P_2_O_5_ (0 g·kg^−1^), K_2_O (0.45 g·kg^−1^), sheep manure (6 g·kg^−1^) and Pb^2+^ (350 mg·kg^−1^); (4) Based on the soil fertility characteristics of Urumqi, the recommended optimal fertilizer application rate was: high phosphorus fertilizer P_2_O_5_ (495 kg·hm^−2^), low-level potassium fertilizer K_2_O (330 kg·hm^−2^), medium-level nitrogen fertilizer N (660 kg·hm^−2^) (or low-level nitrogen fertilizer N (330 kg·hm^−2^) + high-level organic manure sheep manure (19,800 kg·hm^−2^), which can achieve high yield while ensuring that the Pb content in the fruits does not exceed the standard. Strengthening control of effective and reasonable fertilization methods in Urumqi agricultural land is helpful to reduce the Pb content in vegetables.

## 1. Introduction

Soil is the basis for plant growth, and the soil-plant system is the basic structural unit of the biosphere, which provides strong productivity for human beings. Soil is often limited by a low level of available nutrients, so fertilization is often necessary to replenish them. But, fertilization can introduce heavy metals into the soil [1]. Excessive use of fertilizers can lead to a decrease in soil pH, increase the availability of natural heavy metal lead (Pb) in soil, cause harm to soil physical properties, and then affect plant growth and development [2]. Lead from anthropogenic sources typically builds up mostly in the top layer of soil, and as depth increases, its concentration declines and only limited amounts of the lead in soil are thought to be soluble and hence available for plant uptake because of its high binding [3]. Due to the detrimental effects on soil microbiology, food safety and crop growth, lead accumulation in soils is a severe concern in agricultural production [4]. Lead’s speciation in soil has a significant impact on its bioavailability and, consequently, its toxicity to plants and microorganisms [5]. Studies have found that even short-term exposure of plants to the toxic effects of the heavy metal lead can lead to a significant reduction in their microbial activity; to counteract these effects, many plants and microorganisms have evolved detoxification mechanisms using organic colloidal components [6,7,8].

Nitrogen fertilizer is the most widely used fertilizer in agricultural production and plant cultivation, and can directly affect plant yield and quality. Through the root box test, it was found that ammonium nitrogen fertilizer could reduce the pH value of the foundation soil, improve the activity of Pb in the rhizosphere soil, and promote the absorption of Pb by maize, while the effect of nitrate nitrogen is the opposite [9]. Phosphorus is the second macronutrient necessary for plant growth. Existing studies indicate that phosphorus compounds could reduce the absorption of Pb by plants by enhancing the adsorption and complexation of soil colloid surfaces, on the other hand, the application of ammonium phosphorus fertilizer and calcium-phosphorus fertilizer would reduce soil pH value, and the decrease of pH value would increase the solubility and mobility of heavy metals, improve their bioavailability and toxicity, and promote the absorption of heavy metals by plants [10]. Potassium is also one of the essential nutrient elements for plant growth, mainly used to improve crop quality [11]. The research showed that K^+^ could compete with Pb^2+^ for adsorption sites on the surface of soil particles to increase the content of water-soluble and exchangeable Pb, thus increasing the content of its available state [12,13]. Organic fertilizer can not only improve soil fertility but also affect the form of heavy metals in soil and their absorption by plants [14,15]. Studies have shown that both the available Pb content and wheat absorption were significantly negatively correlated with the application of organic fertilizer, while the available Pb content in soil was significantly positively correlated with the amount of Pb absorbed by wheat [15,16]. The application of organic fertilizer could inhibit the absorption of Pb in soil by wheat [17]. Therefore, it is of great significance for the safe and sustainable development of agricultural products to understand the effects of nitrogen, phosphorus, potassium and organic fertilizers on the content of heavy metals in soil and plants [18,19].

Urumqi is the most distant inland city from the ocean in the world, with an average altitude of 680~920 m. It has a semi-arid continental climate with a large temperature difference between day and night, abundant solar and thermal resources, and its terrain and soil types are suitable for growing vegetables, rice and other crops [20]. The research showed that the average contents of Cd, Pb and Hg in farmland soil in Urumqi exceed the soil background value in Urumqi, of which the Pb content (24.83 mg·kg^−1^) is about the soil background value (17.95 mg·kg^−1^) 1.4 times, which has not yet exceeded the limit requirements in the “Evaluation Standards for Environmental Quality of Edible Agricultural Products Origin” (HJ 332-2006) (Pb ≤ 50 mg·kg^−1^), and is far lower than the “Soil Environmental Quality” (GB 15618-2018) The control standards (6.5 < pH ≤ 7.5, Pb ≤ 120 mg·kg^−1^) but there have been occurrences of Pb exceeding the standard in locally produced vegetables in Urumqi [21]. The previous data of our research group showed that when the lead content in the soil of Urumqi did not exceed the standard, improper fertilization would lead to excessive lead in the edible part of the radish. Therefore, exploring the effect of fertilization on the growth and Pb content of vegetables in Urumqi City can provide the theoretical basis for safe and reasonable fertilization. In this study, the common solanaceous vegetable pepper was taken as the object, and the effects of different levels of nitrogen, phosphorus and potassium fertilizer and organic fertilizer on the growth and Pb content of pepper under Pb stress were studied by pot orthogonal experiment in the shed [22].

## 2. Materials and Methods

### 2.1. Test Materials

The pepper variety tested was the improved pig large intestine (PLI) produced by Xi an Qinshu Agriculture Company(Xi’an, Shaanxi Province, China). The tested soil was chernozem soil purchased from Mingzhu Flower Market, Urumqi City (The soil comes from the soil of agricultural land around Urumqi City and is mainly used for planting fruits and vegetables), and the organic fertilizer was pure sheep dung fermented at high temperature from Bafang Zehui Company(Xi’an, Shaanxi Province, China). The basic physical and chemical properties of soil and organic manure (sheep manure) were shown in Table 1. The nitrogen fertilizer used in the experiment was urea (N 46%), the phosphate fertilizer was calcium triple superphosphate (P_2_O_5_ 52%), the potassium fertilizer was potassium sulphate (K_2_O 54%), and Pb was added as an external compound Pb(NO_3_)_2_ (containing Pb^2+^ 62%).

### 2.2. Experimental Design

An orthogonal experiment L_16_4^5^ (5 factors, 4 levels, a total of 16 treatments) was used to design and analyze the effects of the combined action of four fertilizers and Pb^2+^ on the Pb absorption of PLI. The five factors were N, P_2_0_5_, K_2_O, sheep manure and Pb^2+^. The four levels are blank, low, medium and high. The experiment is done by adding fertilizers and compounds to the soil. The five influencing factors in the experiment were added to the soil in the form of fertilizers and compounds. The orthogonal experimental design is shown in Table 2.

In this experiment, potted plants were used. In order to avoid the interference of rainwater, the experiment was carried out in the greenhouse behind the Agricultural Building of Xinjiang Agricultural University. The seedlings were raised on 7 April 2022, and a total of 48 flower pots were treated in 16 groups (3 repetitions in each group) on 14 May. 7 kg of soil were respectively loaded into the pots, and the corresponding fertilizer and Pb(NO_3_)_2_ of each treatment were evenly mixed into the soil according to Table 2 (the fertilizer was applied in the form of base fertilizer (60%) + top fertilizer (40%), and Pb(NO_3_)_2_ was applied at one time during the application of base fertilizer), and 1 L of distilled water was poured and left to stand. On 25 May, the PLI seedlings were transplanted into each flower pot, and one PLI was transplanted in each flower pot. Irrigation was carried out with distilled water. The irrigation amount was 1 L/pot every 5 days, and a total of 22 irrigations were carried out until harvest. The PLI received uniform natural light during the growing period and was harvested on 9 September 2022. Soil pH was measured after the PLI was harvested (Soil pH was extracted using a soil-water ratio of 2.5:1 and determined by a pH meter (Mettler Toledo FE28-Standard)).

### 2.3. Measurements and Methods

After harvesting the PLI, the plant height of the PLI was measured with a tape measure and the number of fruits was recorded. Then the PLI was divided into four parts: root, stem, leaf and fruit. The samples were rinsed and washed with deionized water 2–3 times, the surface moisture was dried, and the samples were weighed fresh. The plant samples were then dried at 105 °C for 30 min, dried in an oven at around 65 °C, ground and sieved, and stored for later use.

The total Pb content of plant and soil samples was determined by System 5000 graphite furnace atomic absorption spectrometer. The content of total nitrogen was determined by Kjeldahl method, the content of total phosphorus was determined by molybdenum-antimony colorimetric method, and the content of total potassium was determined by flame photometry. The organic matter content was heated by potassium bichromate. The available phosphorus content was determined by sodium bicarbonate extraction and molybdenum-antimony sulfate resistance colorimetric method. The content of available potassium was determined by ammonium acetate extraction and flame photometer method.

### 2.4. Data Analysis

Microsoft Excel 2010, IBM SPSS Statistics 25.0 and Orthogonal Design Assistant II v3.1 were used for statistical analysis of the data. Pb content was calculated as fresh weight, with Mean ± SD value. Plot with Origin 2018 and Sigmaplot 10.0 software.

## 3. Results

### 3.1. Effects of Fertilization on Plant Height and Fruit Weight of PLI under Pb Stress

As can be seen from Figure 1, The highest PLI plant height in the 8th group was 49.87 cm, while the lowest PLI plant height in the 11th group was 27.83 cm; the highest number of PLI fruits in the 2nd group was 20, and the lowest number of PLI fruits in the 13th group was 7; the highest PLI fruit weight in the 12th group was 165.47 g, and the lowest PLI fruit weight in the 13th group was 67.50 g. Most of the soil pH was weakly acidic except for groups 1, 3 and 4.

According to Table 3, the factors affecting the plant height of PLI in order of importance were: A (N) > B (P_2_O_5_) > C (K_2_O) > E (Pb^2+^) > D (sheep manure), and N fertilizer has the greatest influence on PLI plant height. It can be seen from the K value that the level combination of factors that promote PLI plant height to reach the maximum is: A2 B4 C2 D1 E3, namely N (0.15 g·kg^−1^), P_2_O_5_ (0.225 g·kg^−1^), K_2_O (0.15 g·kg^−1^), sheep manure (0 g·kg^−1^), Pb^2+^ (100 mg·kg^−1^); The level combination of factors contributing to the minimum plant height of PLI was as follows: A3 B3 C4 D3 E4, namely N (0.3 g·kg^−1^), P_2_O_5_ (0.15 g·kg^−1^), K_2_O (0.45 g·kg^−1^), sheep manure (6 g·kg^−1^), Pb^2+^ (350 mg·kg^−1^).

The main and secondary influencing factors of different fertilizer applications on PLI fruit weight were as follows: B (P_2_O_5_) > D (sheep manure) > A (N)> C (K_2_O) > E (Pb^2+^), phosphorus fertilizer had the greatest effect on PLI fruit weight. The horizontal combination of factors that promote the maximum fruit weight of PLI is: A2 B4 C2 D4 E3, namely N (0.15 g·kg^−1^), P_2_O_5_ (0.225 g·kg^−1^), K_2_O (0.15 g·kg^−1^), sheep manure (9 g·kg^−1^), Pb^2+^ (100 mg·kg^−1^); The horizontal combinations of factors that promoted PLI fruit weight to reach the minimum value were: A4 B1 C4 D2 E4, namely N (0.45 g·kg^−1^), P_2_O_5_ (0 g·kg^−1^), K_2_O (0.45 g·kg^−1^), sheep manure (3 g·kg^−1^), Pb^2+^ (350 mg·kg^−1^).

As can be seen from Figure 2, nitrogen fertilizer was positively correlated with PLI plant height, phosphorus fertilizer was significantly positively correlated with PLI fruit weight (*p* < 0.05), and nitrogen fertilizer was significantly negatively correlated with soil pH value (*p* < 0.05).

### 3.2. Analysis of Pb Content in PLI

The results of Pb content determination in PLI were shown in Table 4. As can be seen from Table 4, the Pb content range of different parts of PLI in each treatment was generally in the order of root (0.104~1.086 mg·kg^−1^) > stem (0.032~0.298 mg·kg^−1^) > leaf (0.022~0.072 mg·kg^−1^), indicating that Pb was mainly concentrated in the root. Pb content in fruits ranged from 0.011 mg·kg^−1^ to 0.085 mg·kg^−1^, among which the highest Pb content in fruits treated 16 was 0.085 mg·kg^−1^, but it did not exceed the limit (0.1 mg·kg^−1^) in the “National Standard for Food Safety-Limit of Pollutants in Food” (GB 2762-2017).

The variance analysis of Pb content in different parts of PLI applied with different kinds of fertilizer is shown in Figure 3. The difference in Pb content in different parts of PLI under different treatments shows that there is no significant difference in Pb content in different parts of PLI in groups 1, 7, 15 and 16. Pb content in the roots and stems of PLI in groups 3, 5 and 11 was significantly different from that in the leaves and fruits. The Pb content in the roots of PLI in groups 2 and 6 was significantly different from that in the leaves and fruits. Pb content in the roots of PLI seedlings in groups 4 and 13 was significantly different from that in stems, leaves and fruits. The Pb content in the roots of group 14 was significantly higher than that in stems, leaves and fruits, and the Pb content in stems, leaves and fruits also showed significant differences, The content of Pb in roots was significantly different from that in stems, leaves and fruits.

In addition, the difference between groups in the same part of PLI under different treatments was as follows: the Pb content of group 14 in the root was significantly different from that of group 7 and group 15; Pb content in the stems of group 11 was significantly different from that in the stems of group 4. Pb content in leaf group 7 was significantly different from that in groups 1, 2, 3, 5, 6, 8, 9, 10, 11, 12, 13 and 15. Pb content in fruit group 7 was significantly different from that in groups 2, 3, 8, 9, 12, 13, 14, 15 and 16. 

### 3.3. Effects of Fertilization on Pb Content in PLI under Pb Stress

According to the R-value in Table 5, the main and secondary influencing factors of different fertilizers on Pb content in PLI roots were as follows: E (Pb^2+^) > A (N) > C (K_2_O) > B (P_2_O_5_) > D (sheep manure), and Pb^2+^ had the greatest influence on Pb content in PLI roots. It can be seen from the K value that the combination of factors that promote the maximum Pb content in PLI root is as follows: A4 B1 C4 D1 E4, namely N (0.45 g·kg^−1^), P_2_O_5_ (0 g·kg^−1^), K_2_O (0.45 g·kg^−1^), sheep manure (0 g·kg^−1^), Pb^2+^ (350 mg·kg^−1^); The level combination of factors to promote the minimum Pb content in PLI root was as follows: A3 B3 C1 D3 E1, namely N (0.3 g·kg^−1^), P_2_O_5_ (0.15 g·kg^−1^), K_2_O (0 g·kg^−1^), sheep manure (6 g·kg^−1^), Pb^2+^ (0 mg·kg^−1^).

The main and secondary influencing factors on Pb content in PLI fruit were as follows: E (Pb^2+^) > C (K_2_O) > A (N) = D (sheep manure) > B (P_2_O_5_), and Pb^2+^ had the greatest influence on Pb content in PLI fruit. The combination of factors that promote the maximum Pb content of PLI fruit is as follows: A2 B1 C4 D3 E4, namely N (0.15 g·kg^−1^), P_2_O_5_ (0 g·kg^−1^), K_2_O (0.45 g·kg^−1^), sheep manure (6 g·kg^−1^), Pb^2+^ (350 mg·kg^−1^); The combination of factors that promote the minimum Pb content of PLI fruit is as follows: A4 B4 C3 D1 E3, namely N (0.45 g·kg^−1^), P_2_O_5_ (0.225 g·kg^−1^), K_2_O (0.3 g·kg^−1^), sheep manure (0 g·kg^−1^), Pb^2+^ (100 mg·kg^−1^).

### 3.4. Effect of Fertilization Level on Pb Content in PLI under Pb Stress

The correlation analysis of various indicators of soil and PLI is shown in Figure 4. The Pb content of PLI roots is significantly positively correlated with the Pb content of leaves, and the Pb content of PLI stems and fruits is significantly positively correlated. The ratio of PLI stem to root and shoot showed a significant negative correlation; soil Pb content was significantly positively correlated with plant height, and plant height was significantly positively correlated with fruit weight.

The linear relationship between Pb content in different parts of PLI and different fertilizer applications is shown in Table 6. The variation trend of lead content in each part of PLI is shown in Figure 5. According to Table 6 and Figure 5, we can find that there was a significant negative correlation between sheep manure and Pb content in PLI stems (r =−0.990, *p* = 0.010). Pb^2+^ was positively correlated with Pb content in root (r = 0.993, *p* = 0.007), and Pb content in leaf (r = 0.979, *p* = 0.021). There was no significant correlation between Pb content in different parts of PLI and other fertilizers.

## 4. Discussion

Fertilization is one of the important agricultural measures to ensure the increase of agricultural production and income, and it also affects the adsorption and resolution of heavy metals in soil, the physicochemical properties of rhizosphere soil and the absorption of heavy metals by crops [23]. Pb is a non-essential element for plants, thus it is hazardous even in low quantities. It easily passes from the soil and atmosphere to plants [2]. 

Nitrogen is an essential element for plant growth and is contained in the composition of vitamins and energy systems in plants [24]. In this experiment, there was a positive correlation between nitrogen fertilizer and plant height of PLI, that is, the increase of nitrogen fertilizer would promote the growth of plant height. The study showed that the average growth rate and plant height of plants increased when the nitrogen supply of plants increased, which was consistent with the results of this experiment. However, nitrogen application is considered to be the main driving factor of soil acidification, because the input of nitrogen fertilizer will make the nitrogen in the soil easy to nitrate, and the generated NO_3_^−^ will produce H^+^, which is easy to leach out of the soil with base ions [25]. The study showed that the nitrification of nitrogen in the soil would produce a large number of protons and lead to a decrease in soil pH, while the addition of fertilizer nitrogen in the soil would lead to more significant soil acidification [25]. In this experiment, there was a significant negative correlation between nitrogen fertilizer and soil pH value, that is, the increase of nitrogen fertilizer led to the decrease of soil pH value. The decrease in soil pH can increase the content of extractable heavy metal elements in soil, improve the bioavailability of heavy metals, and enable plants to enrich more heavy metals [26]. Therefore, attention should be paid to the decrease in soil pH value caused by nitrogen fertilizer application.

Rational application of phosphorus fertilizer can increase crop yield, improve crop quality, promote flowering and fruit of jacket vegetables, and improve results. The research showed that after applying phosphorus fertilizer, the yield of PLI increased with the increase of phosphorus application, and the yield was the highest at the highest phosphorus application [18,27]. In this experiment, there was a significant positive correlation between phosphorus fertilizer and PLI fruit weight, that is, the increase of phosphorus fertilizer would increase PLI fruit weight, possibly because phosphorus would participate in the metabolism and transportation of PLI carbohydrates, which is conducive to the growth of PLI fruit.

Organic fertilizer can not only improve soil fertility but also affect the form of heavy metals in soil and their absorption by plants [15,28]. In this experiment, the content of organic fertilizer was negatively correlated with the Pb content in the roots of PLI and significantly negatively correlated with the Pb content in the stems. Pb intake studies in plants revealed that roots have the ability to absorb large amounts of Pb while limiting their translocation to higher parts of the plant [9,29]. This may be because organic fertilizer has a fixed effect, including the adsorption of heavy metals by macromolecules of solid organic matter and clay minerals in the soil, limiting its mobility and reducing the availability of heavy metals [30]. Moreover, after entering the root, ions are transported to the xylem through both ectoplasmic and symplast pathways, and then to the aboveground part. However, the low permeability of the ectoplasmic barrier to heavy metal ions makes it difficult for Pb to transfer to the stem of PLI [2].

In this experiment, the Pb content in the roots of group 14 was significantly higher than that in stems, leaves and fruits, and the Pb content in stems, leaves and fruits also showed significant differences, which may be because high levels of Pb^2+^ were enriched in PLI roots through cation adsorption, oxidation and reduction reactions of exchange complexes and because heavy metals in roots were not easily transported upward from stems. Pb content in the stems of group 11 was significantly different from that in the stems of group 4, which may be due to the application of high-concentration organic fertilizer, which resulted in the chelation of Pb^2+^ with organic matter and other metal oxides, which restricted its mobility and reduced the absorption and transport capacity of Pb in the stems of PLI. Pb content in fruit group 7 was significantly different from that in groups 2, 3, 8, 9, 12, 13, 14, 15 and 16, which may be because potassium plays an important role in plant energy metabolism, acting as a cofactor or activator of many enzymes in carbohydrate and protein metabolism. In addition to helping the roots to absorb Pb from the soil and transfer it from the stem to the leaves, it can also accelerate the process of Pb absorption by the leaf surface itself.

According to the statistics of soil census data, the total nitrogen content of cultivated soil in Urumqi city was 1.13 g·kg^−1^, the total phosphorus content was above 0.8 g·kg^−1^ and the total potassium content was about 19.28g·kg^−1^, showing the phenomenon of nitrogen deficiency, phosphorus deficiency and potassium enrichment [31]. Pb content in soil is relatively low, generally < 100 mg·kg^−1^.In view of the fact that the Pb content of all PLI fruits treated in this experiment is not beyond the standard, the optimal fertilizer application can be recommended according to the factor level combination (A2 B4 C2 D4) that promotes the maximum fruit weight of PLI: N (330 kg·hm^−2^), P_2_O_5_ (495 kg·hm^−2^), K_2_O (330 kg·hm^−2^), sheep manure (19,800 kg·hm^−2^); Or according to the combination of treatment 12 with the maximum fruit weight (A3 B4 C2 D1), namely N (660 kg·hm^−2^), P_2_O_5_ (495 kg·hm^−2^), K_2_O (330 kg·hm^−2^), sheep manure (0 kg·hm^−2^), both of which were treated with high-level phosphorus fertilizer and low-level potassium fertilizer. medium level of nitrogen fertilizer (or low level of nitrogen fertilizer + high level of organic fertilizer) can achieve a higher yield and not exceed the standard of Pb in fruit.

## 5. Conclusions

(1) The Pb content in PLI fruit ranged from 0.011 mg·kg^−1^ to 0.085 mg·kg^−1^, which did not exceed the limit (0.1 mg·kg^−1^) in the National Standard for Food Safety-Limit of Pollutants in Food (GB 2762-2017).

(2) The effect order of fertilization on fruit weight of PLI was P_2_O_5_ > sheep manure > N > K_2_O > Pb^2+^; The horizontal combination of factors that promoted PLI fruit weight to reach the maximum value were N (0.15 g·kg^−1^), P_2_O_5_ (0.225 g·kg^−1^), K_2_O (0.15 g·kg^−1^), sheep manure (9 g·kg^−1^), Pb^2+^ (100 mg·kg^−1^).

(3) The effects of fertilization on Pb content in PLI fruit were as follows: Pb^2+^ > K_2_O > N = sheep manure > P_2_O_5_; The level combination of factors that promoted the maximum Pb content in PLI fruit was N (0.15 g·kg^−1^), P_2_O_5_ (0 g·kg^−1^), K_2_O (0.45 g·kg^−1^), sheep manure (6 g·kg^−1^), Pb^2+^ (350 mg·kg^−1^). There was a significant negative correlation between sheep manure and Pb content in PLI stem (r = −0.990, *p* = 0.010). There was a significant positive correlation between Pb^2+^ and Pb content in the root (r = 0.993, *p* = 0.007) and in the leaf (r = 0.979, *p* = 0.021). There was no significant correlation between Pb content in different parts of PLI and other fertilizers.

(4) Recommended optimal fertilizer application: high-level phosphorus fertilizer P_2_O_5_ (495 kg·hm^−2^), low-level potassium fertilizer K_2_O (330 kg·hm^−2^), medium-level nitrogen fertilizer N (660 kg·hm^−2^) (or low-level nitrogen fertilizer N (330 kg·hm^−2^) + high-level organic manure sheep manure (19,800 kg·hm^−2^), Can achieve high yield and fruit Pb does not exceed the standard.

## Figures and Tables

**Figure 1 plants-12-02960-f001:**
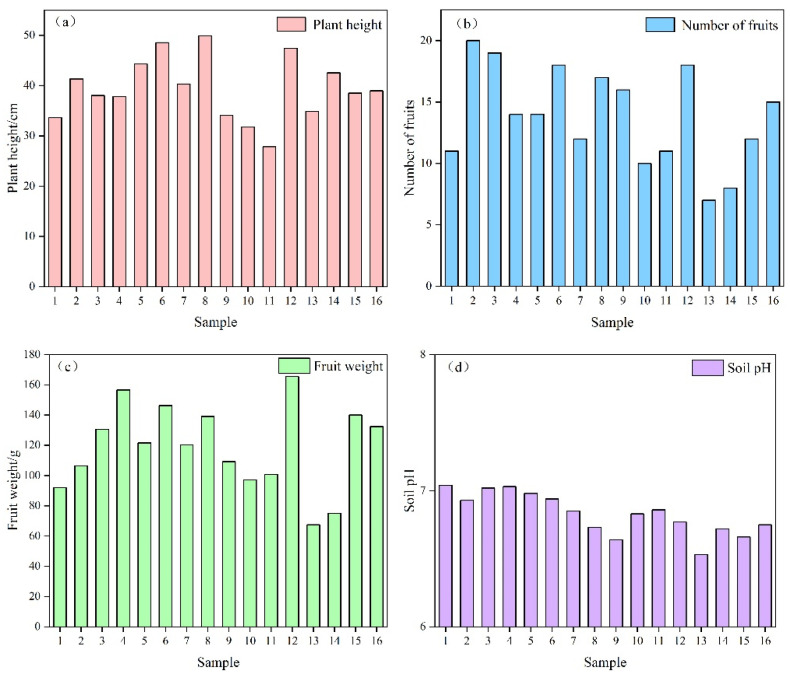
PLI growth indexes and soil pH table of each group. (**a**) shows the data of pepper plant height under different treatments, (**b**) shows the data of pepper fruit number under different treatments, (**c**) shows the data of pepper fruit weight under different treatments, (**d**) shows the data of soil pH under different treatments.

**Figure 2 plants-12-02960-f002:**
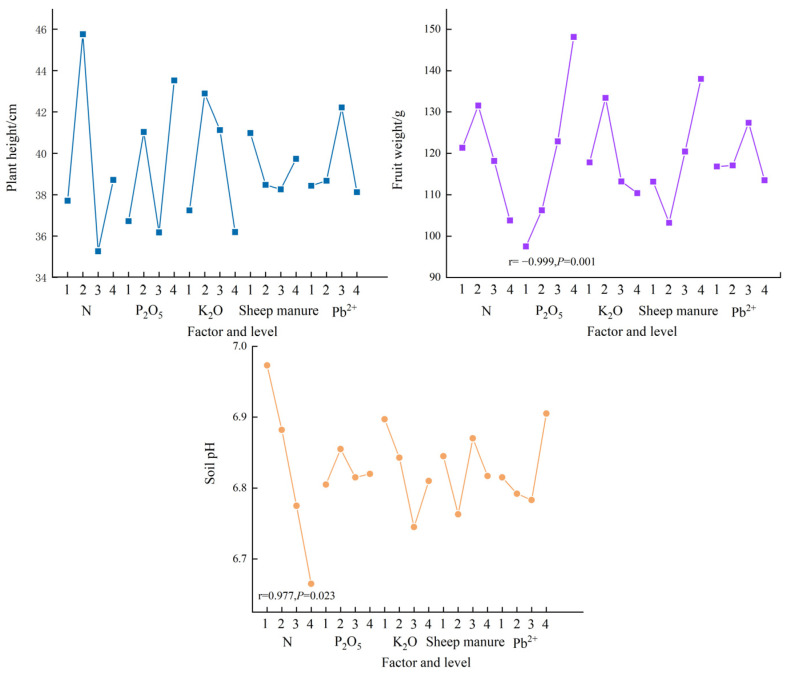
Effects of fertilization on PLI plant height, fruit weight and soil pH. Note: R values represent correlation, *p* values represent significance.

**Figure 3 plants-12-02960-f003:**
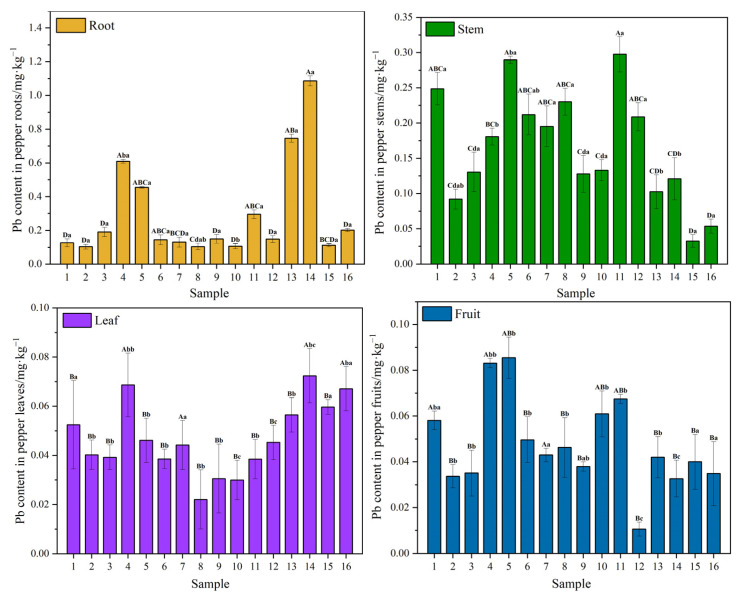
Lead content and variance analysis of PLI in different parts and under different treatments. Note: Different capital letters indicate that at the *p* < 0.05 level, Pb content in the same PLI parts under different treatments is significantly different; different lowercase letters indicate that at the *p* < 0.05 level, Pb content in different PLI parts under the same treatment is significantly different.

**Figure 4 plants-12-02960-f004:**
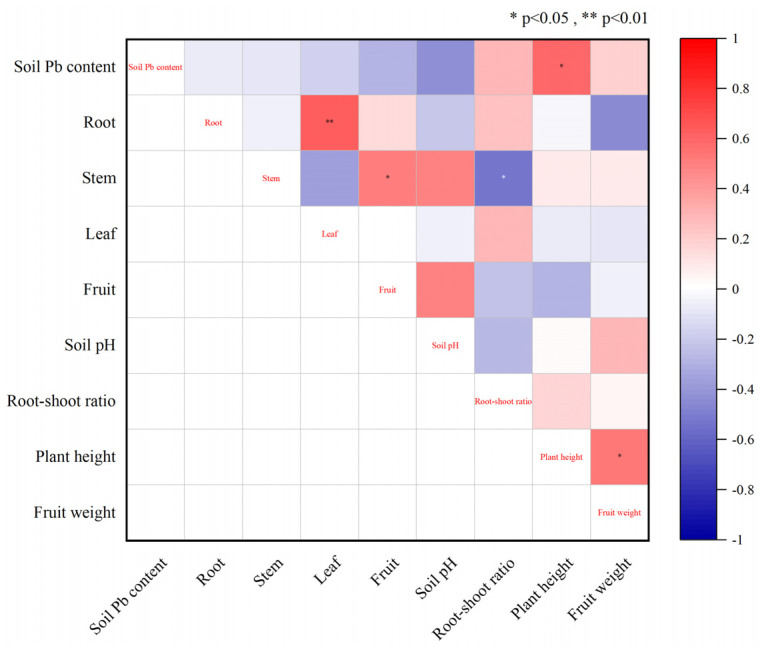
Correlation analysis of various indicators between soil and PLI. Note: The figure represents the Pb content in each part of the root, stem, leaf and fruit.

**Figure 5 plants-12-02960-f005:**
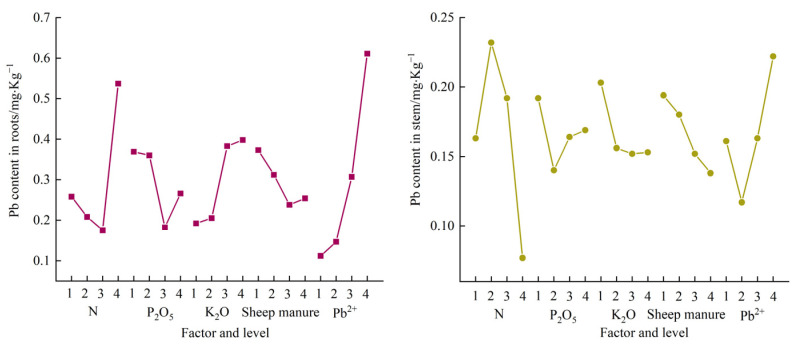
Effects of fertilization on Pb content in PLI under Pb stress.

**Table 1 plants-12-02960-t001:** Basic physicochemical properties of soil and organic fertilizer (sheep manure).

Sample	Total nitrogen	Total phosphorus	Total potassium	Available nitrogen	Available phosphorus
	/g·kg^−1^	/g·kg^−1^	/g·kg^−1^	/mg·kg^−1^	/mg·kg^−1^
Soil	9.36	1.84	1.87	80.91	42.16
Organic fertilizer	12.79	2.392	2.73	95.37	61.96
Sample	Available potassium	Organic matter	Conductivity	Pb	pH
	/g·kg^−1^	/g·kg^−1^	/ms·cm^−1^	/mg·kg^−1^	
Soil	0.11	127.58	2.37	15.74	7.21
Organic fertilizer	0.18	227.36	1.75	2.19	7.09

**Table 2 plants-12-02960-t002:** Orthogonal experimental design.

Sample	Factor Level					Concentration				
	A	B	C	D	E	N	P_2_O_5_	K_2_O	Sheep Manure	Pb^2+^
	N	P_2_O_5_	K_2_O	Sheep Manure	Pb^2+^	/g·kg^−1^	/g·kg^−1^	/g·kg^−1^	/g·kg^−1^	/mg·kg^−1^
1	1	1	1	1	1	0	0	0	0	0
2	1	2	2	2	2	0	0.075	0.15	3	30
3	1	3	3	3	3	0	0.150	0.30	6	100
4	1	4	4	4	4	0	0.225	0.45	9	350
5	2	1	2	3	4	0.15	0	0.15	6	350
6	2	2	1	4	3	0.15	0.075	0	9	100
7	2	3	4	1	2	0.15	0.150	0.45	0	30
8	2	4	3	2	1	0.15	0.225	0.30	3	0
9	3	1	3	4	2	0.30	0	0.30	9	30
10	3	2	4	3	1	0.30	0.075	0.45	6	0
11	3	3	1	2	4	0.30	0.150	0	3	350
12	3	4	2	1	3	0.30	0.225	0.15	0	100
13	4	1	4	2	3	0.45	0	0.45	3	100
14	4	2	3	1	4	0.45	0.075	0.30	0	350
15	4	3	2	4	1	0.45	0.150	0.15	9	0
16	4	4	1	3	2	0.45	0.225	0	6	30

**Table 3 plants-12-02960-t003:** Range analysis of fertilization on PLI plant height, fruit weight and soil pH.

Sample	Parameter	A	B	C	D	E
		N	P_2_O_5_	K_2_O	Sheep Manure	Pb^2+^
Plant height	K1	37.708	36.717	37.242	40.983	38.433
	K2	45.758	41.033	42.892	38.475	38.675
	K3	35.267	36.175	41.125	38.258	42.216
	K4	38.717	43.525	36.191	39.733	38.125
	R	10.491	7.350	6.701	2.725	4.091
Main and secondary order: A(N) > B(P_2_O_5_) > C(K_2_O) > E(Pb^2+^) > D(Sheep manure)						
Highest level A2 B4 C2 D1 E3						
Minimum level A3 B3 C4 D3 E4						
Fruit weight	K1	121.355	97.530	117.808	113.173	116.813
	K2	131.550	106.237	133.402	103.215	117.077
	K3	118.155	122.913	113.222	120.438	127.420
	K4	103.767	148.148	110.395	138.002	113.518
	R	27.783	50.618	23.007	34.787	13.902
Main and secondary order: B(P_2_O_5_) > D(Sheep manure) > A(N) > C(K_2_O) > E(Pb^2+^)						
Highest level A2 B4 C2 D4 E3						
Minimum level A4 B1 C4 D2 E4						
Soil pH	K1	6.973	6.805	6.897	6.845	6.815
	K2	6.882	6.855	6.843	6.763	6.792
	K3	6.775	6.815	6.745	6.870	6.783
	K4	6.665	6.820	6.810	6.817	6.905
	R	0.308	0.050	0.152	0.107	0.122
Main and secondary order: A(N) > C(K_2_O) > E(Pb^2+^) > D(Sheep manure) > B(P_2_O_5_)						
Highest level A1 B2 C1 D3 E4						
Minimum level A4 B1 C3 D2 E3						

Note: K1, K2, K3 and K4 are the sum of the indicators at each level of each factor, K1 represents the sum of the values of the test indicators corresponding to the “1” level. R is called the range, and the largest K is subtracted from the smallest K.

**Table 4 plants-12-02960-t004:** Pb content in PLI (fresh weight).

Sample	Pb^2+^ Content/mg·kg^−1^			
	Root	Stem	Leaf	Fruit
1	0.126 ± 0.014	0.249 ± 0.023	0.052 ± 0.018	0.058 ± 0.004
2	0.104 ± 0.021	0.092 ± 0.014	0.040 ± 0.006	0.034 ± 0.005
3	0.191 ± 0.002	0.130 ± 0.028	0.039 ± 0.005	0.035 ± 0.010
4	0.610 ± 0.017	0.181 ± 0.012	0.069 ± 0.013	0.083 ± 0.002
5	0.455 ± 0.026	0.290 ± 0.005	0.046 ± 0.009	0.085 ± 0.009
6	0.144 ± 0.018	0.212 ± 0.029	0.039 ± 0.004	0.050 ± 0.010
7	0.131 ± 0.036	0.195 ± 0.029	0.044 ± 0.010	0.043 ± 0.003
8	0.104 ± 0.001	0.230 ± 0.019	0.022 ± 0.012	0.046 ± 0.013
9	0.149 ± 0.038	0.128 ± 0.026	0.031 ± 0.014	0.038 ± 0.002
10	0.106 ± 0.004	0.133 ± 0.015	0.030 ± 0.008	0.061 ± 0.010
11	0.295 ± 0.000	0.298 ± 0.025	0.038 ± 0.008	0.067 ± 0.002
12	0.148 ± 0.022	0.209 ± 0.020	0.045 ± 0.007	0.011 ± 0.003
13	0.746 ± 0.103	0.102 ± 0.024	0.056 ± 0.007	0.042 ± 0.009
14	1.086 ± 0.022	0.121 ± 0.030	0.072 ± 0.011	0.033 ± 0.008
15	0.112 ± 0.033	0.032 ± 0.009	0.060 ± 0.003	0.040 ± 0.012
16	0.202 ± 0.012	0.054 ± 0.010	0.067 ± 0.009	0.035 ± 0.014
Min~Max	0.104~1.086	0.032~0.298	0.022~0.072	0.011~0.085

**Table 5 plants-12-02960-t005:** Range analysis table of the effect of fertilization on Pb content in PLI under Pb stress.

Sample	Parameter	A	B	C	D	E
		N	P_2_O_5_	K_2_O	Sheep Manure	Pb^2+^
Root	K1	0.258	0.369	0.192	0.373	0.112
	K2	0.208	0.360	0.205	0.312	0.147
	K3	0.175	0.182	0.383	0.238	0.307
	K4	0.537	0.266	0.398	0.254	0.611
	R	0.362	0.187	0.206	0.135	0.499
Main and secondary order: E(Pb^2+^) > A(N) > C(K_2_O) > B(P_2_O_5_) > D(Sheep manure)						
Highest level A4 B1 C4 D1 E4						
Minimum level A3 B3 C1 D3 E1						
Stem	K1	0.163	0.192	0.203	0.194	0.161
	K2	0.232	0.140	0.156	0.180	0.117
	K3	0.192	0.164	0.152	0.152	0.163
	K4	0.077	0.169	0.153	0.138	0.222
	R	0.155	0.052	0.051	0.056	0.105
Main and secondary order: A(N) > E(Pb^2+^) > D(Sheep manure) > B(P_2_O_5_) > C(K_2_O)						
Highest level A2 B1 C1 D1 E4						
Minimum level A4 B2 C3 D4 E2						
Leaf	K1	0.050	0.046	0.049	0.053	0.041
	K2	0.038	0.045	0.048	0.039	0.045
	K3	0.036	0.045	0.041	0.045	0.045
	K4	0.064	0.051	0.050	0.050	0.056
	R	0.028	0.006	0.009	0.014	0.015
Main and secondary order: A(N) > E(Pb^2+^) > D(Sheep manure) > C(K_2_O) > B(P_2_O_5_)						
Highest level A4 B4 C4 D1 E4						
Minimum level A3 B2 C3 D2 E1						
Fruit	K1	0.053	0.056	0.053	0.036	0.051
	K2	0.056	0.045	0.043	0.047	0.037
	K3	0.044	0.046	0.038	0.054	0.035
	K4	0.038	0.044	0.057	0.053	0.067
	R	0.018	0.012	0.019	0.018	0.032
Main and secondary order: E(Pb^2+^) > C(K_2_O) > A(N) = D(Sheep manure) > B(P_2_O_5_)						
Highest level A2 B1 C4 D3 E4						
Minimum level A4 B4 C3 D1 E3						

Note: K1, K2, K3 and K4 are the sum of the indicators at each level of each factor, K1 represents the sum of the values of the test indicators corresponding to the “1” level.; R is called the range, and the largest K is subtracted from the smallest K.

**Table 6 plants-12-02960-t006:** Linear relationship between fertilization and Pb content in PLI under Pb stress.

Sample	Root		Stem		Leaf		Fruit	
	r	*p*	r	*p*	r	*p*	r	*p*
N	0.628	0.372	−0.585	0.415	0.400	0.600	−0.891	0.109
P_2_O_5_	−0.713	0.287	−0.272	0.728	0.674	0.326	−0.813	0.187
K_2_O	0.925	0.075	−0.804	0.196	−0.126	0.874	0.103	0.897
Sheep manure	−0.907	0.093	−0.990	0.010 **	−0.063	0.937	0.906	0.094
Pb^2+^	0.993	0.007 **	0.874	0.126	0.979	0.021 *	0.744	0.256

Note: “*”, *p* < 0.05, “**”, *p* < 0.01.

## Data Availability

Unable to obtain data due to privacy concerns.

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
