# Peer review of "Influence of Fertilization on Growth and Lead Content of Pepper under Lead Stress"

_plants, 2023, doi:10.3390/plants12162960_

Round 1

Reviewer 1 Report

Title

Integrated fertilization” - what did you mean by that - complex?

Abstract

pepper was planted in L1645 pot orthogonal experiment” - such abbreviated designations do not mean much to the reader, please briefly describe the studied variants.

Keywords: Repeating the same terms in the title and keywords is not recommended. You can choose others from your manuscript - such as Latin name of pepper, sheep manure, NPK fertilizers, fruit weight, plant height, etc.

Introduction

Line 33.” Excessive uses of fertilizers increase the availability of naturally occurring heavy metals lead (Pb) in the form of fertilizers and pesticides…” – unclear, rephrase.

Line 45. “According to Hemida” - what does it mean?

Line 48. “detoxifying processes. with organic and colloidal components.” - ??

Line 51.” Lou conducted…” – what is Lou? Is it a reference? Reference Nr. 9 – another authors.

Line 54. “Phosphorus fertilizer is the second macronutrient necessary for plant growth.” - fertilizer is not a macronutrient, be careful what you write.

Line 55. “Seaman found… “– Seaman et al.,   a reference is required at this point. The same for “Basta showed…” for Line 57.

Line 61. “Potash fertilizer is also one of the essential nutrient elements” - fertilizer is not a macronutrient, be careful what you write.

Line 62. “The research by Tu et al…..” - why 2 references if one of them is not Tu et al.

Line 75. What is Urumgi? A small introduction to the research site is necessary - country, geographical location, soils, pollution problems….

Line 75. What is Wang, what is 3414 fertilization test?

Line 78. What is real estate vegetables?

Line 83. Why is there (aim of the study) a reference? Has someone else done this research before, other than you? So why are you authors? I really don't understand.

Materials and Methods

2.1. Test materials

It is not clear what variety was used in the study. It is also recommended to give the pepper Latin name.

Is the typical soil of Urumgi, where vegetables are usually grown, similar to chernozem soil? Is such a substrate chosen for the convenience of conducting experiments? Describe the soils of Urumgi, at least roughly. Some explanation here would be necessary. Did you take into account that the P fertilizer also contained K, and the lead nitrate – nitrogen?

Table 1. What method was used for determination of available forms of N, P, K? About Pb – is it total or available concentration? About pH – in water or KCl or CaCl2, etc.

2.2. Experimental design.

Table 2. Concentrations in soil?

Line 105. “Seedlings will be raised” - if it's already done, then was.

Line 106. What is basin and what is pot. Are they synonyms?

 It is necessary to clearly indicate how much soil was in the pot, how many repetitions for each variant. Did each variant have 1 pot?

2.3. Measurements and methods

Line 118. “The plant samples were then killed” – why so dramatic – were fixed.

Line 122. “basic physical and chemical properties of soil were determined” – what properties were determined.

Nothing is mentioned about plant height measurements, fruit counting.

3. Results and discussion

Since there is no Discussion in this section, it is a Results section.

3.1. Effects of fertilization on plant height and fruit weight of pepper under Pb stress

Figure 1. No standard deviations or significant differences are indicated. There are no pH table, as mentioned. In the Figure, it is not possible to see that the soil pH for variants 1, 3 and 4 are different from the others. However, the pH scale in the Figure does not have to start from zero!

Table 3. What is parameter K1, K2, K3, K4 and R? Nothing described.  Figures and Tables should be self-explanatory.

Figure 2. This Figure contains the same information as Table 3, it is a repetition.

Line 152-157. What is it - the legend for Figure 2, or the text or Table 2 - I can't understand anything.

In general, it is not necessary to repeat in the text what we also see in the table - it makes the text unnecessarily long.

Line 160. “it can be seen from the mean level K value” – what is mean level K value?

3.2. Analysis of Pb content in pepper

Table 4. It is not specified what is significantly different with either letters or asterisks. Figure 3 - the same, but with statistics.

Legend for Figure 3. “Figure 3. Analysis of variance between pepper parts and treatments”. In this Figure we can see Pb concentrations!! No information is given on the system for reflecting the significant differences between treatments.

Line 182. Why capsicum is used?

Line 191. “A large number of studies” – there are no any reference. In general, there is no room for discussion and analysis of the results in the Results section. It should be moved to Discussion.

3.3 Effects of fertilization on Pb content in pepper under Pb stress

Table 5. The same comments as for Table 3. What is K what is R?

Too long, there is no need to retell everything seen in the table.

3.4. Effect of fertilization level on Pb content in pepper under Pb stress

Figure 4. Correlation analysis of various physical and chemical indicators between soil and pepper.

Nothing described – what is roots, leaves, fruits in the Figure? What correlates? Pb content? Where are the various physical and chemical indicators?

Line 249-256. It is a part of Discussion! Did you consider that the applied Pb source also contained the nutrient N?

Table 6. Why extremely significant? Significant at the 0.01 level.

Figure 5. This Figure contains the same information as Table 5, it is a repetition. Why is the concentration of Pb in pepper stems given twice?

4. Conclusions or Discussion

I think it is Discussion.

Line 275. “In this experiment, there was a significant positive correlation between nitrogen fertilizer and plant height of pepper” – where we can see data about significant positive correlation? Please give these data - correlation coefficients in the text or as a Table..

Why is there a reference at the end of the sentence - aren't these your results?!

Line 277. “The Werf study…” – what is Werf study?

Line 280. “nitrogen application is considered to be the main driving factor of soil acidification” – which forms of N fertilizer are most responsible for soil acidification – urea, ammonium, nitrate?

Line 283. Wu showed – what is Wu?

Line 285. In this experiment, there was a significant negative correlation between nitrogen fertilizer and soil pH value? Where we can see data about significant positive correlation? Please give these data - correlation coefficients.

Line 293. Zhang showed that – what is Zang?

Line 297. In this experiment, there was a significant positive correlation between phosphorus fertilizer and pepper fruit weight - where we can see data about significant positive correlation? Please give these data - correlation coefficients.

Line 302-304. Why is there a reference at the end of the sentence - did Qin et al really do the research for you and already published the data? This is absurd!!

Line 306-311. Too few references.

Line 309. “Moreover, after entering the root, ions in the soil are transported to” - doesn't make sense, after entering the root, these ions are no longer in the soil.

Line 315. Can we even talk about the deficiency or abundance of nutrients based on their total content, which is not available to plants?

5. Conclusions

Since this is a practice-oriented study, it would be recommended in the Conclusions, if any specific doses of fertilizers are given, to indicate them also in units that are applicable in agriculture – on m2 or per ha. Hardly anyone knows how much his soil weighs. Are these recommended fertilizer rates the same or different from what is currently commonly used for vegetables in your region? Look wider and more generally, not just at specific grams, but at levels.

General conclusions

The article is devoted to such a topical, important and interesting issue as the role of fertilization in growing plants in conditions of heavy metal pollution. It is always important to get a high yield, but at the same time safe for food. However, there are so many inaccuracies in the article. The description of the methodology should be definitely improved. The Results section is too long with a lot of repetitions. Duplications in the presentation of data in the Results section should be removed. More attention should be paid to the correct presentation of tables and figures so that the results can be clearly understood. The discussion part should be based on correct references. The reference section must be supplemented.

I recommend accepting this article in Plants after major revision. Efforts should be made to improve the scientific quality of this work, as the data material obtained in this study could be of interest to both plant physiology and agricultural scientists.

Author Response

Thank you for your letter and for the reviewers’ comments on our manuscript entitled “Influence of integrated fertilization on growth and lead content of pepper under lead stress”. Those comments are very helpful for improving our manuscript, as well as the important guiding significance to our research. We have made revisions which are marked in red in the manuscript according to the comments.Please see the attachment.

Reviewer 2 Report

Dear Authors,

The subject of the study is interesting and topical, with scientific and practical importance.

The introduction is presented correctly, in accordance with the subject. Numerous scientific articles, in concordance to the topic of the study, were consulted.

Methodology of the study was clearly presented, and appropriate to the proposed objectives.

The obtained results are important and have been analyzed and interpreted correctly, in accordance with the current methodology.

The discussions are appropriate, in the context of the results, and was conducted compared to other studies in the field.

The scientific literature, to which the reporting was made, is recent and representative in the field.

Some suggestions and corrections were made in the article.

The following aspects are brought to the attention of the authors.

1.

Page 1, row 8

“L1645” What does it represent?

The explanation is given in parentheses on page 3, line 97.

Please analyze whether the explanation in the parenthesis would be appropriate here as well (it is the first use).

2.

The order of the chapters in the article

Please see the recommended order for the chapters in the content of the article, according to Instructions for Authors, and Microsoft Word template, Plants journal.

By redoing the chapters, the order of citation of the bibliographic sources in the text will be changed, and it will be necessary to harmonize with the content of the References chapter.

3.

Space between the word and the parentheses in which the bibliographic sources are cited

e.g.

page 1, row 32

“soil [1].” Instead of “soil[1].”

Please check the entire article and correct as necessary

4.

Table caption

e.g.

Page 6, rows 152 – 157

“Table 2. …”

What does this text represent?

Should it be on another page, in another location?

Please check and make the necessary correction.

5.

Page 9, row 242

It is recommended that a subchapter starts with text and not directly with a figure (Figure 4).

Moreover, it is necessary to refer to a figure in the text before its presentation.

Please analyze and fix this aspect.

6.

It is necessary to refer to the Figures and Tables in the text before their presentation.

e.g. page 10, Table 6, Figure 5.

Please review and fix this issue.

7.

Page 11.

"Discussion" instead of "Conclusions"

8.

References

If the order of the chapters is redone, according to the Instructions for Authors, and Microsoft Word template, Plants journal, the order of the bibliographic sources may change.

It is recommended to pay attention so that the order of the bibliographic sources is consistent.

Author Response

Thank you for your comments on our manuscript entitled “Influence of integrated fertilization on growth and lead content of pepper under lead stress”. Those comments are very helpful for improving our manuscript, as well as the important guiding significance to our research. We have made revisions which are marked in red in the manuscript according to the comments. Please see the attachmen.

Round 2

Reviewer 1 Report

General recommendations and questions

Although the authors have significantly improved the content of the article, there are still many gaps and unclear questions.

Keywords are generally unchanged.

Introduction

Line 50-53: unclear, rephrase.

Materials and Methods

2.1. Test materials

It is recommended to give the pepper Latin name.

My previous questions: Is the typical soil of Urumgi, where vegetables are usually grown, similar to chernozem soil? Is such a substrate chosen for the convenience of conducting experiments? Describe the soils of Urumgi, at least roughly. Some explanation here would be necessary -  You have answered me in the reply letter, but not in the text of the article.

Critically serious question that has not been answered - Did you take into account that the P fertilizer also contained K, and the lead nitrate – nitrogen?

If we can still ignore the N content in lead nitrate, because it is not high, then K in potassium dihydrogen phosphate is more than phosphorus. You used it as a phosphorus fertilizer. I repeat the question - has it been taken into account that it is also K fertilizer? This significantly affects the trial scheme. Explain!

Table 1. What method was used for determination of available forms of N, P, K? About Pb – is it total or available concentration? About pH – in water or KCl or CaCl2, etc.

You have answered something to me in the reply letter, but you have not changed anything in the text of the article. This is not correct at all. At least the reference or extraction method should be specified! The answer – “I used the conventional method of soil agrochemical analysis” - not valid for Q1 scientific journal.

2.2. Experimental design.

Table 2. Concentrations in soil? Not added to the table.

2.3. Measurements and methods

Line 122. “basic physical and chemical properties of soil were determined” – what properties were determined; what methods were used? It is not enough “Using the conventional method of soil agrochemical analysis to determine the basic physical and chemical indexes of samples”. Reference or explanation is necessary. It is not difficult or complicated to do.

3. Results

3.1. Effects of fertilization on plant height and fruit weight of pepper under Pb stress

Figure 1. In the Figure, it is still not possible to see that the soil pH for variants 1, 3 and 4 are different from the others. Why is the pH scale not changed, starting from, for example, 5 or 6?

Table 3. What is parameter K1, K2, K3, K4? It is necessary to explain precisely what is the parameter K1, K2, K3, K4 in the Figures. You write the sum of the values. The sum of what indicators? Looks rather mean value???

3.2. Analysis of Pb content in pepper

Legend for Figure 3. “Figure 3. Variance analysis of Pb content between pepper parts and treatment”. In this Figure we can see Pb concentrations and statistically significant differences between treatments, but not variance analysis.

In general, there is no room for discussion and analysis of the results in the Results section. It should be moved to Discussion – for example line 202-203, 209-211, 214-218.

3.4. Effect of fertilization level on Pb content in pepper under Pb stress

Line 247. The correlation analysis of various physical and chemical indicators of soil and pepper is shown in Figure 4. Where are the various physical and chemical indicators of soil? Only soil pH results!

Table 6. Why extremely significant? This is not a correct wording in the scientific literature. High level of significance, significant at the 0.01 level.

Discussion.

Line 278-281. “In this experiment, there was a significant positive correlation between nitrogen fertilizer and plant height of pepper” – where we can see data about significant positive correlation? Please indicate these data - correlation coefficients and/or significance level simply in the text, even if in parentheses.

Your answer in Response to reviews: “As can be seen from this figure, N fertilizer is positively correlated with plant height.”

This is not serious! I can only see a trend, but no significant correlation. There are criteria for statistical significance. Besides, you explain something again in the Response letter, but not in the text.

Line 289. In this experiment, there was a significant negative correlation between nitrogen fertilizer and soil pH value? Please give these data - correlation coefficients or significance level.

Line 299. In this experiment, there was a significant positive correlation between phosphorus fertilizer and pepper fruit weight - where we can see data about significant positive correlation? Answer in the Response letter - as can be seen from this figure (Figure 2), there is a significant positive correlation between P-fertilizer and fruit weight is not sufficient. Please give these data - correlation coefficients or significance level.

5. Conclusions

Previous question: since this is a practice-oriented study, it would be recommended in the Conclusions, if any specific doses of fertilizers are given, to indicate them also in units that are applicable in agriculture – on m2 or per ha. Hardly anyone knows how much his soil weighs.

Your answer: The design of this experiment is that the weight of each pot of pepper soil is 7 kg, and the fertilizer amount in the medium level is similar to that in Urumqi agricultural land, but the low level is different from the high level.

What is the meaning of this answer? Have you seriously considered it? If it was an anecdote, I would laugh, but now I want to cry.

What would change if there were 9 kg of soil in the pot instead of 7 kg? Who really doesn't know that the low level is different from the high level.

Let's consider a simple example: simple calculation - 1 m2 is 200 L of soil at a depth of 0-20 cm. Assuming that 1 L is about 1 kg, that's 200 kg of soil. 1 ha - about 2000000 kg of soil. So you recommend 60 g N per 1 m2 or 600 kg N per ha. If you know the bulk density of the soil, the calculation will be more accurate.

General conclusions

The article is devoted to such a topical, important and interesting issue as the role of fertilization in growing plants in conditions of heavy metal pollution. However, even after the corrections, the article still contains quite a lot of inaccuracies. The description of the methodology should be definitely improved. Main question – fertilizers used in the experiment! It should be clearly understood whether there is no joint influence of P and K in the variant with P fertilization. More attention should be paid to the correct presentation of tables and figures so that the results can be clearly understood. Clarifications and corrections should be made in the text of the manuscript, not just written to the reviewer. I recommend accepting this article in Plants after major revision.

Round 3

Reviewer 1 Report

The authors have put in the work to ensure that the manuscript can be published in Plants after revision.